# AMM15: A new high resolution NEMO configuration for operational simulation of the European North West Shelf

Jennifer A. Graham[1-2], Enda O'Dea[1], Jason Holt[3], Jeff Polton[3], Helene T. Hewitt[1], Rachel Furner[1], Karen Guihou[3-4], Ashley Brereton[3], Alex Arnold[1], Sarah Wakelin[3], Juan Manuel Castillo Sanchez[1], and C. Gabriela Mayorga Adame[3]

[1]Met Office, FitzRoy Road, Exeter, UK, EX1 3PB
[2]Now at Centre for Environment, Fisheries and Aquaculture Science, Pakefield Road, Lowestoft, UK, NR33 0HT
[3]National Oceanography Centre, Liverpool, UK, L3 5DA
[4]Departamento Oceanografía, Servicio de Hidrografía Naval, Buenos Aires, Argentina

*Correspondence to:* Jennifer A. Graham (jennifer.graham@cefas.co.uk)

**Abstract.** This paper describes the next generation ocean forecast model for the European North West Shelf, which will become the basis of operational forecasts in 2018. This new system will provide a step change in resolution, and therefore our ability to represent small scale processes. The new model has a resolution of 1.5 km, compared with a grid spacing of 7 km in the current operational system. AMM15 (Atlantic Margin Model, 1.5 km) is introduced as a new regional configuration of

NEMO v3.6. Here we describe the technical details behind this configuration, with modifications appropriate for the new high resolution domain. Results from a 30 year non-assimilative run, using the AMM15 domain, demonstrate the ability of this model to represent the mean state and variability of the region.

Overall, there is an improvement in the representation of the mean state across the region, suggesting similar improvements may be seen in the future operational system. However, the reduction in seasonal bias is greater off-shelf than on-shelf. In

the North Sea, biases are largely unchanged. Since there has been no change to the vertical resolution or parameterisation schemes, performance improvements are not expected in regions where stratification is dominated by vertical processes, rather than advection. This highlights the fact that increased horizontal resolution will not lead to domain-wide improvements. Further work is needed to target bias reduction across the North West Shelf region.

## 1   Introduction

The Met Office runs an operational ocean forecast for the European North West Shelf (NWS). This system is developed by both the Met Office and National Oceanography Centre, through the Joint Weather and Climate Research Programme. The current operational capabilities for the NWS are at a resolution of 7 km (O'Dea et al., 2017). While this configuration is able to reproduce the large-scale circulation across the shelf, it fails to resolve a host of dynamical features, such as mesoscale eddies, frontal jets, internal tides, and tidally rectified transport (e.g., Holt et al., 2017). All of these features make a substantial

contribution to the fine scale currents and material distribution throughout the shelf seas. For example, mesoscale eddies can

have a radius $< 10\,\text{km}$ on mid-latitude continental shelves, and are crucial in transporting heat, freshwater and nutrients in the region (e.g., Badin et al., 2009). To simulate these processes in numerical models, we therefore require higher resolution.

Across the NWS, the majority of previous high resolution studies ($< 2\,\text{km}$ grid spacing) have been limited to shelf regions (e.g., Holt and Proctor, 2008). These studies have shown the impact of resolution, for example resolving buoyancy-driven currents along tidal mixing fronts (Holt and Proctor, 2008), and cross-front transfer through baroclinic instabilities (Badin et al., 2009). However, using a purely on-shelf domain, these studies neglect the potential influence of shelf-break dynamics.

A recent study by Guihou et al. (2017) has demonstrated the potential impact of increased resolution across the NWS, using a domain that extends to $\sim 20°\text{W}$, comparable to the existing forecast system (O'Dea et al., 2017). With a resolution of $\sim 1.8\,\text{km}$, internal waves are generated along the shelf break, as well as locally around bathymetric features on the shelf, such as sea mounts. Resolving such features has significant impacts on vertical mixing and stratification across the shelf, and therefore they need to be represented to make accurate ocean forecasts across the region.

The next generation ocean forecast model for the European NWS is introduced here, with the intention that it will become operational in 2018. The new configuration has a resolution of $1.5\,\text{km}$ throughout the NWS domain. This will allow a step-change in our simulations, with the aim of improved representation of spatial and temporal variability. This configuration will typically be used to produce forecasts on time scales of hours to weeks. Surface products are made available on hourly to sub-hourly timescales (e.g. temperature, salinity, velocity and surface height), with full-depth products also available on hourly, daily and monthly timescales. The full operational system will include data analysis. This system may also then be used to produce decadal reanalysis products, similar to that produced for the existing operational domain (O'Dea et al., 2017).

Before the inclusion of data analysis, it is important to understand any underlying biases in the free-running model, along with potential model drift. Here we present a 30 year non-assimilative run, using the new high resolution domain. This long simulation demonstrates the ability of this model to represent the mean state and variability of the region. The existing operational system has known biases, outlined in O'Dea et al. (2017). We compare the results from this new simulation with the performance of the current system, to illustrate where there is likely to be the greatest improvements. Hereafter, the new $1.5\,\text{km}$ domain will be referred to as AMM15 (Atlantic Margin Model, $1.5\,\text{km}$ resolution). The existing operational model will be referred to as AMM7 ($7\,\text{km}$ resolution).

## 2   Model Development

### 2.1   Core Model Description

AMM15 is a regional configuration of NEMO (Nucleus for European Models of the Ocean), at version 3.6 stable (Madec, 2016). Compared with the current operational system (AMM7), this configuration has a new domain, at higher resolution (Figure 1). However, aside from the horizontal grid, AMM15 shares many features with the previous configuration, which has been described in O'Dea et al. (2012, 2017). Here we outline some of the key components and parameterizations. The horizontal resolution is sufficient for resolving the internal Rossby radius on the shelf, which is of order $4\,\text{km}$ (Holt and Proctor, 2008). As such, only a minimal amount of eddy viscosity is applied in the lateral diffusion scheme, to ensure model stability.

For momentum and tracers, bi-laplacian viscosities are applied on model levels, using coefficients of $6 \times 10^7 \, \mathrm{m^4 \, s^{-1}}$ and $1 \times 10^5 \, \mathrm{m^4 \, s^{-1}}$, respectively.

Tides are the dominant source of variability across the majority of the North West Shelf. A non linear free surface is therefore implemented using the variable volume layer (Levier et al., 2007). Time splitting is included, with a barotropic time step chosen automatically to satisfy a maximum Courant number of 0.8. For a baroclinic time step of 60 seconds there are then 17 barotropic time steps for each baroclinic.

The vertical coordinate system is based on a $z* - \sigma$ approach, as described in Siddorn and Furner (2013). The stretching function used here allows for more uniform surface heat fluxes across the domain, with the thickness of the surface cell set to $\leq$ 1 m. With terrain-following coordinates, large slopes between adjacent grid cells can lead to pressure gradient errors. To reduce such errors, vertical cells can be masked over slopes which exceed a specified value, $r_{max}$ (where $r = (h_i - h_{i+1})/(h_i + h_{i+1})$, and $h_{i,i+1}$ are adjacent bathymetry points). Terrain following coordinates are fitted to a smoothed envelope bathymetry, with the level of smoothing based on the chosen $r_{max}$ value. In regions where the smoothed model levels become deeper than the input bathymetry, these levels are then masked. The $r_{max}$ value was here chosen to be 0.1. This is a lower value than used in previous configurations. However, with increased resolution, the model bathymetry is rougher, resolving steeper gradients and canyons along the shelf-break. This value was then chosen to ensure stability in the configuration, without the need to smooth the input bathymetry.

For AMM15, there is no increase in the vertical resolution, using 51 vertical levels. The vertical parameterizations in AMM15 then remain similar to the current operational system. The Generic Length Scale scheme is used to calculate turbulent viscosities and diffusivities (Umlauf and Burchard, 2003). Surface wave mixing is parameterized by Craig and Banner (1994). A minimum surface roughness is specified as 0.02 m. Dissipation under stable stratification is limited using the Galperin limit (Galperin et al., 1988) of 0.267 (Holt and Umlauf, 2008). Bottom friction is controlled through a log layer with a non-linear drag coefficient set at 0.0025.

## 2.2 Domain and Bathymetry

The domain for AMM15 has a smaller area than the current operational domain (Figure 1). This is due to the computational demands of higher resolution, considering both ocean-only as well as future coupled simulations. The model domain extends from approximately $45°$N to $63°$N, with a uniform grid spacing of $\sim 1.5$ km in both the zonal and meridional direction. Compared to AMM7, the number of wet cells has increased by a factor of $\sim 15$. While the operational run time is still uncertain (pending future developments, including data assimilation), the physics only AMM15 configuration requires approximately 400 node hours per day, compared to 20 node hours per day for AMM7. The storage costs have also increased (by a factor of $\sim 11$ for standard daily output files).

The domain boundaries were chosen carefully, to ensure that they would not limit representation of major current pathways, whilst also ensuring that the grid would be compatible with coupled simulations (e.g. considering location of mountain ranges and the Mediterranean within the domain used for ocean-atmosphere coupling) (Lewis et al., 2017). This chosen common domain is now also in use at the Met Office for uncoupled operational UK weather forecasts (extending the existing operational

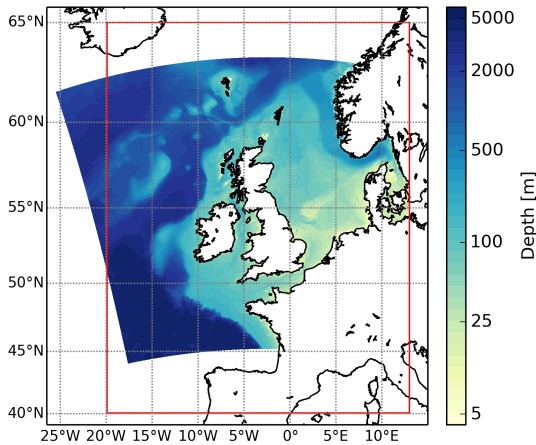

**Figure 1.** Map illustrating the location and bathymetry of the model domain (indicated by the shaded region). Shading shows bathymetry from EMODnet [m] (note logarithmic scale). Red line illustrates the extent of the current operational domain, AMM7 (7 km resolution).

domain, e.g., Tang et al. (2013)). To the south, the AMM15 boundary was chosen far enough north of the Spanish coast, so that the shelf-break transport could flow into the domain perpendicularly through the relaxation zone (rather than parallel to the boundary), while considering placement in relation to the Gironde Estuary. The northern boundary is placed sufficiently north of the Faroe Islands, to allow transport around the islands, but far enough south to not be concerned with the representation of

overflows or transport around Iceland. The representation of overflows is a longstanding known problem in lower resolution global models (e.g., Beckmann and Döscher, 1997; Roberts and Wood, 1997). Given that lower resolution data (O($1/4°$)) will be used as boundary conditions for this regional model, it is advisable to avoid the overflow region with the domain. To the west, the model extends far enough into the Atlantic to allow off-shelf dynamics to develop away from the shelf-break, reducing potential impacts of boundary conditions on shelf-break exchange. To the east, the boundary remains in the Baltic,

similar to previous versions. However, since the increased resolution allows for potentially improved representation of heat and freshwater transport through Danish Straits, the boundary is now placed at $\sim 12°$E, in the Arkona Basin, rather than within the Kattegat, north of the Danish Straits.

     The bathymetry chosen for AMM15 is EMODnet (EMODnet Portal, September 2015 release). This product was the best available at the time, combining all observations from the region. With increased resolution, increased detail can now be

represented in the model's bathymetry. This detail will contribute to improved representation of small scale processes, in particular along the shelf break (e.g., Aslam et al., 2017). For numerical models, the limitation is that the EMODnet product is referenced to lowest astronomical tide (LAT), whereas the model requires bathymetry referenced to mean sea level (MSL). In the deep ocean this is less of a concern, since the range of the tide is negligible compared with the depth of the ocean. However, this difference is crucial when considering the depth along shallow coastal regions where there are large tidal ranges. To apply

an adjustment from LAT to MSL, we have used an estimate of the LAT from a 19 year simulation of the CS3X tidal model (Batstone et al., 2013). For each point, the lowest tidal depth has then been added to the original EMODnet depth.

EMODnet data is provided with a land sea mask based on OpenStreetMap (2014), which has here been interpolated onto the AMM15 grid. EMODnet data is originally obtained at a higher resolution than AMM15. For grid cells of partial land/sea, they were originally set as land if the EMODnet land mask covered $> 50\%$ of the target grid cell. Following this interpolation, the mask was assessed manually to check the representation of narrow channels, estuaries or small islands. This simulation does not include wetting and drying, so the land sea mask is fixed, and a minimum depth is specified for the input bathymetry. Taking into account the large tidal ranges in the Bristol Channel and Gulf of St. Malo, this minimum depth is specified as $10\,\mathrm{m}$. While the tidal range may be smaller in other regions, this domain-wide minimum depth was chosen for simplicity, as well as consistency with previous configurations. Wetting and drying is not available within NEMO vn3.6, however it is currently under development for NEMO vn4.0. This capability will then be a priority for future development of AMM15.

## 2.3 Forcing and Initialisation

The simulation discussed here covers 30 years, starting in 1985. This is a free running simulation, with no data assimilation. During this time, the regional model is forced with lateral ocean boundary conditions, surface atmospheric forcing, river runoff and tidal forcing.

All lateral boundary conditions except the eastern boundary have been taken from a series of global ocean simulations, carried out with the ORCA025 configuration at the Met Office. For 1985-1989, the boundaries used here come from a free running global ocean hindcast (Megann et al., 2014). This same simulation provided the initial temperature and salinity conditions for the AMM15 hindcast, with the model initialised from rest on $1^{st}$ January 1985. For 1990 onwards, the boundary conditions are taken from the Global Seasonal Forecast System (GLOSEA), version 5 (MacLachlan et al., 2015; Jackson et al., 2016), which includes assimilation of both satellite and in situ observations, where available. Analysis of AMM15 will therefore focus on the period of GLOSEA forcing, allowing a 5 year spinup period prior to this date. For the eastern boundary, conditions have been taken from a regional Baltic simulation (Gräwe et al., 2015). This alternative data set was chosen due to the increased resolution ($1/60°$, as opposed to $1/4°$ in the ORCA025/GLOSEA data), in order to resolve flow through the Arkona Basin ($\sim 12°$E).

For operational purposes, alternative boundary conditions will be used for both the Baltic and Atlantic boundaries. For the Atlantic, these will be derived initially from a $1/12°$ configuration of the North Atlantic (NATL12). For the Baltic, boundary forcing will be provided from operational forecast products available through the Copernicus Marine Environmental Monitoring Service. However, neither of these data sets are available over a sufficient time period be used for this long hindcast.

From each of the chosen data sets, the model boundary was forced with 3D temperature and salinity fields, barotropic velocities, and sea surface height (SSH). For SSH, the global data fields were corrected to remove drift from the free-running 1985-1989 simulation, and then ensure that there was no jump between this and the following data sets. Following the same method outlined in O'Dea et al. (2017), an offset was also applied to the global data to ensure that the mean SSH over this domain was approximately zero. For the Baltic boundary, a different offset was applied to ensure that the mean SSH across the

boundary matched what would have been present in the GLOSEA forcing. This maintains the variability present in the Baltic data, but avoids any SSH difference relative to the other boundaries that might result in anomalous transport into or out of the eastern boundary.

Tidal potential is calculated across the domain for 12 constituents. In addition to this, tidal forcing is applied along the lateral boundaries. Forcing has been applied using the Topex Poseidon crossover solution (Egbert and Erofeeva, 2002), TPXO7.2, Atlantic Ocean 2011-ATLAS. For each of the 12 constituents, amplitude and phase (surface height and velocity) was obtained at a resolution of $1/12°$.

River runoff is based predominantly on a daily climatology of gauge data, averaged for 1980-2014. UK data was processed from raw data provided by the Environment Agency, the Scottish Environment Protection Agency, the Rivers Agency (Northern Ireland) and the National River Flow Archive (gauge data were provided by pers. comm from Dr. S. M. van Leeuwen, CEFAS, Lowestoft, UK). For major rivers that were missing from this data set (e.g. along the French and Norwegian coast), data has been provided from an earlier climatology (Young and Holt, 2007; Vorosmarty et al., 1998). For each river point, a daily freshwater flux is specified with the depth dependent on the average ratio of runoff to tidal range (based on estuary classifications discussed in Cameron and Pritchard (1963)). The runoff temperature is assumed to match the local SST, with no temperature data included in the climatology.

Atmospheric forcing is taken from the European Centre for Medium-Range Weather Forecasts (ECMWF) atmospheric reanalysis product, ERA-Interim (Dee et al., 2011). This has a spectral resolution of T255 ($\sim 79$ km).The operational system will make use of the higher resolution ECMWF Numerical Weather Prediction model ($0.125°$ resolution). Forcing is applied using the CORE bulk forcing algorithm (Large and Yeager, 2009), for the full 30 years of the simulation. All variables are applied at 3-hourly intervals. Light attenuation is set to the standard NEMO tri-band scheme (RGB), assuming a constant chlorophyll concentration of $0.05$ mg g$^{-3}$ (Lengaigne et al., 2007).

## 2.4 Summary of differences between AMM7 and AMM15 simulations

For comparison with the existing operational configuration (AMM7), the results from this long hindcast are compared with the AMM7 hindcast discussed in O'Dea et al. (2017). While the construction of these NEMO configurations is similar, there are some differences between the chosen model parameters and boundary conditions. The key differences are outlined here.

The AMM7 hindcast spans 1981-2012, with the method of forcing and initialisation similar to those outlined for AMM15. However, for its forcing and initialisation, AMM7 used earlier versions of both the ORCA025 and GLOSEA configurations, therefore differences can be expected in these forcing products. The free-running ORCA025 simulation was used for initialisation in January 1981, and boundary forcing up to 1990. The remainder of the simulation used boundary conditions from GLOSEA. Although the GLOSEA data comes from an earlier version than that used for AMM15, both versions include data assimilation. Therefore, there should be greater similarity in the boundary conditions for AMM7 and AMM15 from 1990 onwards. Analysis of model climatology will then focus on a common 20 year period in both simulations, 1991-2010.

With 7 km resolution, no attempt was made to model the Danish Straits. The Baltic boundary was placed north of the Straits, with temperature and salinity relaxed to climatology during the CO5 hindcast. No barotropic forcing was applied at this boundary.

In addition to the differing horizontal resolution and spatial coverage between the AMM15 and AMM7 domains as seen in Fig. 1, the source bathymetry for AMM7 is derived from the much coarser North-West Shelf Operational Oceanographic System (NOOS) dataset. Not only are fine scale features missing from the NOOS bathymetry, but there are significant differences in mean depth in some on shelf regions of the North Sea.

The fresh water riverine input also differs. Instead of the climatology used in AMM15, in AMM7 the rivers were based upon the European version of the hydrological model HYdrological Predictions for the Environment (E-HYPE, version 2.1) (Donnelly et al., 2015). Use of this data allows for potential interannual variability in fresh water fluxes, however fresh water biases in areas such as the German Bight in AMM7 have been attributed to large riverine flux from E-HYPE (O'Dea et al., 2017). The mean total freshwater input from E-HYPE v2.1 was found to be $\sim 18\%$ larger than the climatology. This forcing data was then not chosen for AMM15.

The source of the tidal forcing also differs. AMM7 uses tidal forcing derived from a model of the North Atlantic (Flather, 1981) in contrast to TPXO7.2 data utilized in AMM15.

## 3   Model Comparison and Validation

### 3.1   Tidal harmonics

A large proportion of the model performance across the shelf can be determined by tides. Figure 2 shows the co-tidal plot of the M2 constituent for both AMM15 and AMM7. Both models show a very similar pattern, with good agreement in terms of the location of amphidromes across the shelf. There is a slight shift in the position of the amphidrome off the northern Irish coast, towards Scotland. In the English Channel, there is also a slight shift to the west of the Isle of Wight. At both these locations, this coincides with reduced errors in amplitude and phase in AMM15.

The mean bias and route-mean-square error (RMSE) of major constituents, compared with available tide gauge observations (from NOC Marine Data Products and BODC), is presented in Table 1. For the phase of each constituent, the RMSE is reduced in AMM15. The mean bias is reduced for 4 out of the 7 constituents shown. AMM15 amplitudes show less improvement. The RMSE for most constituents is of the same order in both configurations, with the exception of M4. However, both M2 and M4 show an increased mean bias in AMM15, compared to observations. A summary of errors in the semi-major axis of tidal currents is also presented in Table 1 (analysis follows the same method used in Guihou et al. (2017)). Again, the RMSE and bias are found to be of a similar magnitude in the two configurations, but with a slight increase in both M2 and M4. For M2, positive anomalies in surface height can be seen in particular along the east coast of the UK, and on the west coast of England, in the Irish Sea (Figure 2c,d). The increased mean bias can be partly accounted for by the fact that errors are more uniform across the domain. For AMM7, while the RMSE has a similar magnitude to AMM15, compensating errors in both amplitude and phase are found around the UK, reducing the apparent mean bias.

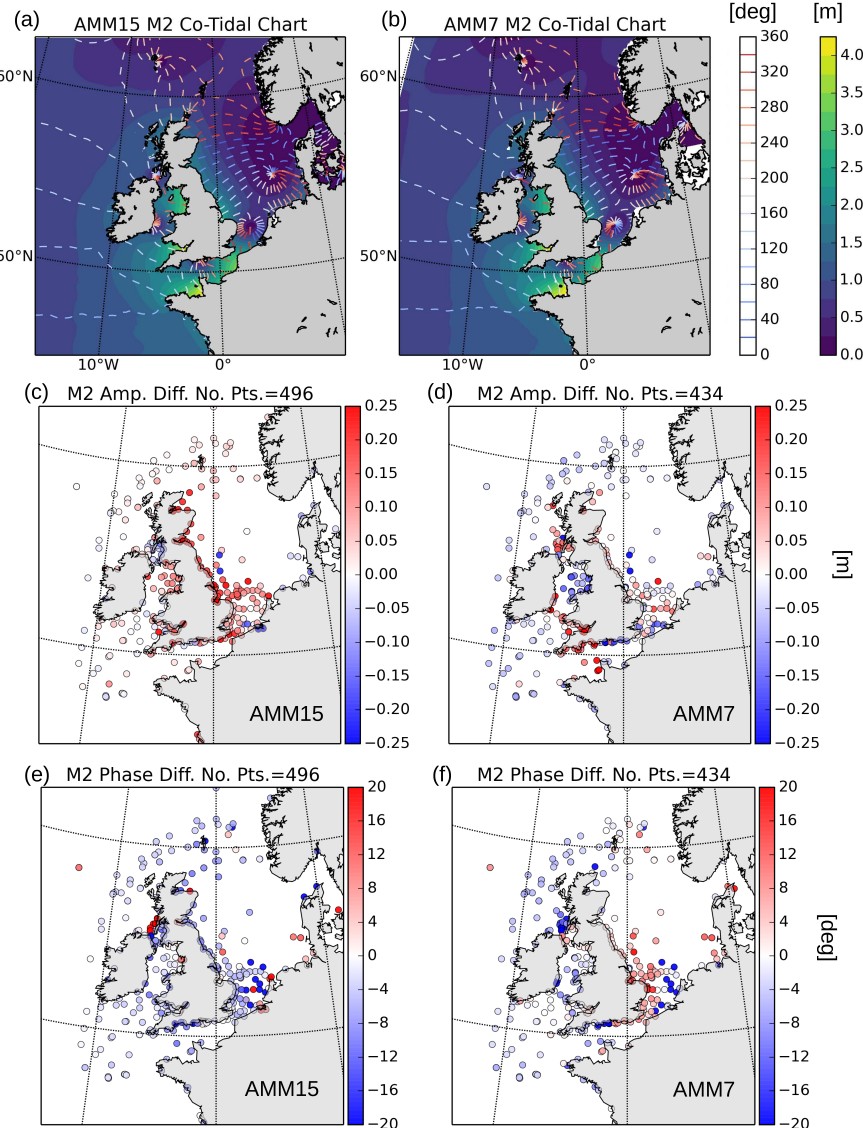

**Figure 2.** Top panels show M2 Co-tidal plots for AMM15 (a) and AMM7 (b). Shading shows M2 amplitude [m]; dashed contours show the phase [deg]. Lower panels show errors in amplitude (c-d) and phase (e-f) for the M2 constituent of the two configurations (model - observations). Observations are tide gauges data from National Oceanography Centre (NOC) Marine Data Products and the British Oceanographic Data Centre (BODC). The number of valid observations (N) is shown for each constituent comparison, depending on the land-sea mask represented in each model configuration. .

While the overall performance of AMM7 and AMM15 are similar (Table 1), anomalies vary across the domain, showing regional improvements. For example, there is particular improvement in the English Channel in AMM15 for both amplitude

**Table 1.** Mean bias and RMS error (model minus observations) for amplitude and phase of major tidal constituents, as well as the semi-major access of tidal currents. Observations are tide gauges data from National Oceanography Centre (NOC) Marine Data Products and the British Oceanographic Data Centre (BODC). Tidal current analysis uses the same data and method outlined in Guihou et al. (2017). The number of valid observations (N) is shown for each constituent comparison, depending on the observed variable and land-sea mask represented in each model configuration.

| Constituent | Amplitude [cm] | | Phase [deg] | | | Current [cm s$^{-1}$] | | |
|---|---|---|---|---|---|---|---|---|
| | RMSE | Bias | RMSE | Bias | N | RMSE | Bias | N |
| AMM15 | | | | | | | | |
| M2 | 12.641 | 6.277 | 10.865 | -3.664 | 496 | 10.307 | 5.368 | 116 |
| S2 | 5.042 | 2.515 | 12.243 | -4.108 | 495 | 3.586 | 1.908 | 116 |
| K1 | 1.820 | 0.836 | 15.102 | -2.361 | 495 | 0.824 | 0.310 | 114 |
| O1 | 1.502 | 0.344 | 13.427 | -2.048 | 494 | 0.747 | 0.160 | 114 |
| N2 | 4.150 | 0.936 | 22.340 | -1.279 | 497 | 2.523 | 0.625 | 112 |
| Q1 | 1.272 | -0.241 | 33.227 | 1.835 | 455 | - | - | - |
| M4 | 8.043 | 3.148 | 59.550 | -10.215 | 460 | 1.525 | 0.230 | 113 |
| AMM7 | | | | | | | | |
| M2 | 11.797 | 0.423 | 12.244 | -1.864 | 434 | 8.895 | 4.094 | 115 |
| S2 | 4.589 | 1.612 | 13.243 | -1.351 | 434 | 3.634 | 1.847 | 115 |
| K1 | 1.642 | 0.538 | 19.933 | -5.051 | 432 | 0.936 | 0.307 | 114 |
| O1 | 1.769 | -0.969 | 23.187 | -2.926 | 434 | 0.621 | -0.182 | 114 |
| N2 | 4.203 | 0.748 | 26.084 | 0.947 | 435 | 2.419 | 0.648 | 112 |
| Q1 | 1.817 | 1.007 | 42.761 | 15.080 | 390 | - | - | - |
| M4 | 4.879 | 0.666 | 84.992 | 12.721 | 395 | 1.224 | -0.033 | 113 |

and phase (figure 2c-f). The amplitude of M2 also has reduced errors off the west coast of Scotland, particularly around the Kintyre Peninsula. There is a considerable difference in the resolution of the coastline between these configurations, which will have a large impact in these regions.

One factor which must be taken into account is that the model applies a minimum depth of 10 m, due to the absence of wetting and drying. The same minimum depth is applied here as in previous configurations. The speed at which the tide travels, and hence the phase of constituents, is dependent on water depth. Hence, while the coastline has been improved, errors are expected due to the depth in shallow coastal regions. This difference in depth will have a large impact in regions such as the East Anglian coast and the Wadden Sea, in the Southern North Sea, as well as shallow estuaries, such as the Bristol Channel, Morecambe Bay and Solway Firth.

There are complex interactions between water depth and the simulation of tidal constituents. The dependency on depth for shallow water wave speed suggests that the simulated speed would be higher with an imposed minimum depth, compared with observations. However, any change in tidal currents will have impacts on the level of bottom friction that is felt, and there

may also be wider impacts on resonance and amplitude across the shelf. Therefore, impacts on tidal circulation are expected to be found downstream of any apparent depth anomalies, as well as more widely across the domain. For AMM15, the M2 constituent shows a negative bias in phase (consistent with increased speed) and positive bias in amplitude (Table 1), with anomalies larger along the east coast of the UK (Figure 2c). Both models show reduced anomalies off-shore, towards the shelf
break, although this reduction appears greater for AMM15 than AMM7.

For AMM7, while there are similar limitations with minimum depth, the coarse coastline may have led to compensating errors in the phase and resonance of tides throughout the region (and hence reduced mean bias). As this configuration has been in operational use for a number of years, the coastline has also been modified to ensure the best possible representation of tides, e.g. deepening or widening channels as required. For AMM15, the initial aim has been to ensure the most realistic coastline
possible. It is therefore encouraging to see that overall there is a comparable if not improved representation of the majority of constituents, despite the considerable differences between both the domains and forcing.

Tests were carried out with a reduced minimum depth of $6\,\mathrm{m}$ in AMM15. However, this led to no significant improvement in tidal simulations. Wetting and drying is currently under development for NEMO vn4.0, with the hope of implementation in future configurations. This would enable 'realistic' depths to be included in the model.

## 3.2  Surface Climatology

Figure 3 shows the mean sea surface temperature (SST) anomalies over the model domain compared with observations, for both AMM15 and the previous operational model. Observations used are a reanalysis version of the Met Office Operational Sea Surface Temperature and Sea Ice Analysis (OSTIA), produced for the European Space Agency's Climate Change Initiative (CCI) (Merchant et al., 2014). This analysis provides a $20\,\mathrm{cm}$ SST product, and is therefore useful for comparing to the upper-
most SST in ocean models. Both models show varying biases during the seasons. Overall the standard deviation of anomalies in AMM15 is reduced compared with AMM7. The largest difference between the two models is found in the north of the domain, where AMM15 is substantially warmer than AMM7, and hence has a reduced cold bias. This cold bias in AMM7 was found to originate from the north western boundary of the domain, near the Iceland coast (O'Dea et al., 2017). The reduction of the cold bias here is then likely related to the change in the location of the boundary. AMM7 has its largest mean SST anomalies
in winter (December, January, February; DJF) and spring (March, April, May; MAM), with a cold bias dominating off-shelf. Analysis of the monthly mean anomalies (not shown) indicates that the cold bias grows progressively during these seasons, reaching a peak in April of $-0.356 \pm 0.643\,°\mathrm{C}$.

Off-shelf, AMM7 was found to alternate between a cold bias in the winter months and warm bias in the summer (June, July, August; JJA) (O'Dea et al., 2017). For AMM15 the model has relatively small bias off-shelf for the majority of the year,
with the exception in JJA when a similar warm bias remains. For AMM15, the largest mean bias occurs in this season, with a mean error of $0.176 \pm 0.304°\mathrm{C}$ (compared with $0.116 \pm 0.331°\mathrm{C}$ for AMM7). This warm bias peaks in July, when there is a mean anomaly of $0.230 \pm 0.334°\mathrm{C}$ across the domain. This bias may in part be related to over-stratification, or limitations of the uniform RGB light attenuation. Both these models use similar vertical mixing schemes, and light attenuation scheme. The choice of light attenuation scheme, and potential impacts on stratification, will be discussed further in Section 3.3.

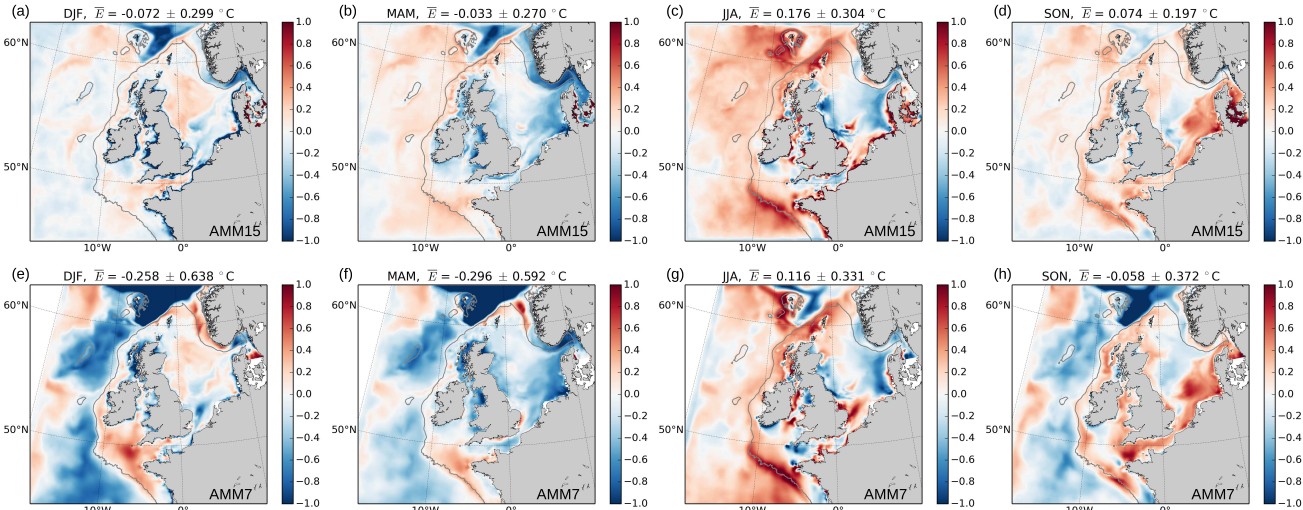

**Figure 3.** Seasonal SST anomalies for model minus observations [°C]. Observations used are OSTIA CCI reanalysis product (Merchant et al., 2014) (NB. OSTIA CCI product only available from Sep 1991). All panels show 20 year-mean anomalies, for period 1991-2010, with anomalies calculated as $\overline{SST_{AMM}} - \overline{SST_{OSTIA}}$. Upper panels (a-d) show anomalies for AMM15-OSTIA, lower panels (e-h) show anomalies for AMM7-OSTIA. Mean errors ($\overline{E}$) and standard deviations are calculated spatially for the region shown (excluding wider AMM7 domain). Grey contour shows the 200 m isobath, to indicate the limit of the continental shelf.

Over the continental shelf break, there is still a warm bias compared with observations during the summer (Figure 3). However, this warm bias has been reduced in AMM15 compared with AMM7. Over the shelf break, the mean SST is typically lower than the surrounding ocean during the summer due to increased vertical mixing. The generation of internal tides at this location provides energy for increased mixing as the internal waves break. This reduces the surface temperature due to mixing with the cooler water beneath the pycnocline. At 1.5 km resolution, internal waves begin to be resolved in the model (as discussed in Guihou et al. (2017)). These processes are not resolved at 7 km resolution. Therefore, AMM15 has increased mixing above the shelf break, contributing to reduced SST in this region. There is still a warm bias in this region, in particular to either side of the shelf break itself. This suggests that AMM15 may not be resolving the full extent of the internal waves, and their impact on vertical mixing.

In the Norwegian Trench, there is a strong cold bias during the spring (Figure 3). In the Baltic, there is a warm bias during Autumn (September, October, November; SON). The anomalies in this region are at times larger than those in AMM7, however there have been significant changes to the Baltic boundary conditions between the two models. Aside from the change in location, AMM15 also has the addition of SSH and barotropic currents forcing at the boundary (where there was none in AMM7). Therefore, we may expect significant changes in transport through the region, which would affect the Norwegian Trench heat and salt transport.

On the shelf, the biases in the two models remain similar. For example, across the North Sea both models show a similar pattern of cool bias during spring-summer, followed by a warm bias in autumn (Figure 3). The warm bias is particularly strong in the southern North Sea, around the German Bight. There are a number of potential causes for these biases. Initially, there could be errors in the surface heat fluxes from ERA-Interim, used to force both simulations. If the assumed SST in ERA-Interim differs to that of OSTIA, then this will be a limiting factor in the ability of the model hindcast to reproduce the observed SST. However, these SST anomalies may also be related to thermal inertia within the ocean, with a lag in the loss or gain of heat through the seasons. Under the same surface heat flux, it will take longer to heat (and cool) a fully mixed water column, than a shallow, stratified surface layer. This may then be related to weak stratification across the shelf. In shallow coastal regions (which are already fully-mixed), the 10 m minimum depth could also be a contributing factor. Another likely source of error is the light attenuation scheme. Across the shelf, the uniform light attenuation will overestimate the depth of light penetration. This may lead to an increase in heat content in the deeper ocean, and hence the ocean will take longer to cool as the mixed layer deepens in the autumn. During spring and early summer, if solar heating is not concentrated within a shallow surface layer (as may occur across a spring chlorophyll bloom), then the heat flux will be distributed with depth and the surface temperature will take longer to increase.

In other coastal regions, anomalies can be found which may be related to over stratification. For much of the British coastline, there are cold anomalies in the winter months, and warm anomalies in the summer. The location of these anomalies is consistent with the location of fresh biases in the surface salinity, which will be discussed below (Figure 4). Further analysis of the stratification in the model will be discussed in the following section (Section 3.3).

Figure 4 shows the surface salinity (SSS) biases, for AMM15 and AMM7 compared with EN4 profiles (Good et al., 2013). There is improvement in the north of the domain, with a reduced fresh bias in AMM15. As discussed in relation to the SST biases, this is likely related to the northern boundary conditions.

One region where AMM15 performs worse than AMM7 is in the Norwegian Trench. There is a fresher anomaly here than in the coarser model. Within the Norwegian Trench, fresh Baltic water is found traveling north on the eastern side. Low salinity is also maintained northwards with the addition of river runoff along the Norwegian coast. On the western side of the Trench, warm saline Atlantic Water flows southward. At the boundary between these two water masses, instabilities and eddies may form, encouraging mixing of properties across the Trench. Previous analysis of AMM7 has shown a dipole across the trench - too fresh along the coast, and too saline off-shore (e.g. Figure 4g). This was believed to be due to a lack of lateral mixing across the Trench. In AMM15, there is no longer a saline bias offshore, consistent with an increased eddy activity in the region. However, there is a stronger fresh bias throughout the trench, extending from the Baltic Sea. This contributes to an increased mean fresh bias over the AMM15 domain.

Further work is needed to attribute this fresh bias within the Norwegian Trench. The Baltic boundary has been altered between the two models, with a significant change in position as well as forcing methods. Such changes would likely have a large impact on the transport into or out of the Baltic. However, the position and forcing along the Atlantic boundaries has also changed, with potential impacts on the balance of transport within the Trench. Further experiments are needed to be able

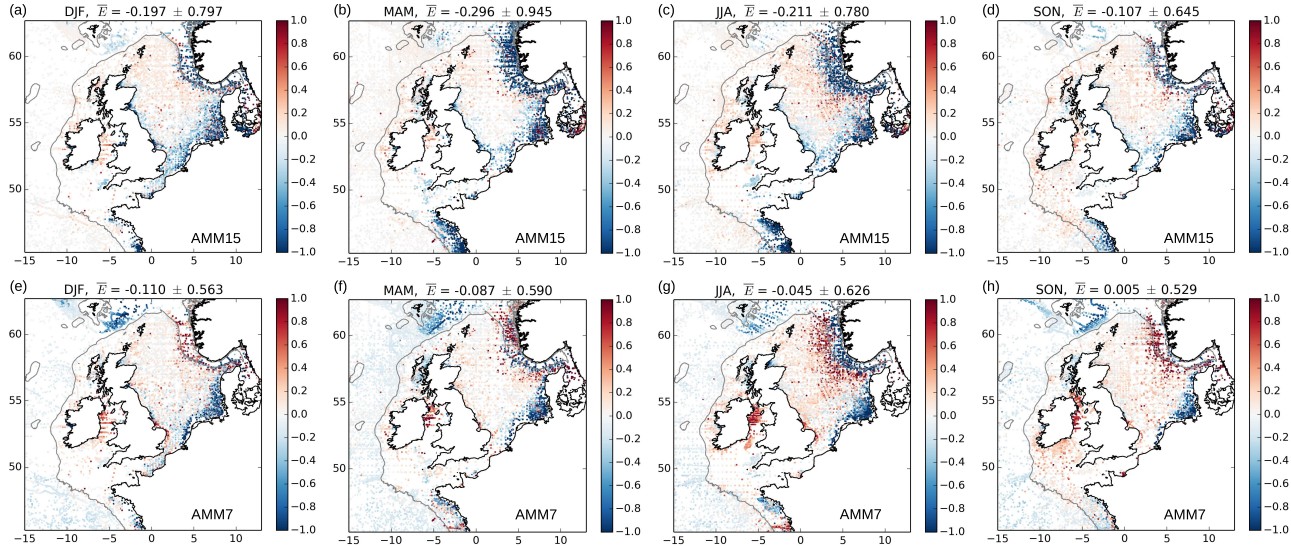

**Figure 4.** Mean seasonal SSS anomalies for model minus observations. Observations used are monthly-mean EN4 profiles (Good et al., 2013). Upper panels (a-d) show anomalies for AMM15-EN4, lower panels show anomalies for AMM7-EN4 (e-h). All panels show monthly anomalies averaged over the period 1991-2010. Mean errors ($\overline{E}$) are calculated for the AMM15 domain region (excluding the wider AMM7 domain). Grey contour shows the 200 m isobath, to indicate the limit of the continental shelf.

to attribute anomalies to either of the new boundary locations or forcing products. Changes in any salinity bias may also be influenced by local river runoff as well as the large scale transport.

Elsewhere there has also been a freshening close to the coast (Figure 4). The river fluxes have been altered between the two models. Overall the climatology has a reduced total freshwater input compared with E-HYPE. However, in some regions, such as along the British and Irish coast, the mean runoff is higher in the climatology (O'Dea et al., 2017). Comparing the conditions in the southern North Sea, AMM15 is fresher than AMM7. However, the sign of anomalies along the coast can vary. In places there is a dipole where AMM15 is fresher at the coast and more saline off shore (Figure 4). This suggests that AMM7 may be more diffusive within river plumes, for example allowing freshwater input from the Rhine to be advected off-shore, whereas AMM15 keeps a narrower plume close to the coast. Indeed, the lateral diffusion prescribed in AMM15 is lower than that used in AMM7, due to the increased resolution and hence ability to resolve mesoscale processes on the shelf (Section 2.1). While 1.5 km is not sufficient to fully resolve plume dynamics, this response is consistent with previous studies on the impact of resolution for plume dynamics (e.g. Bricheno et al., 2014). A similar dipole response can be seen in the SST, indicating a change in stratification in the region, associated with the shift in position of the river plume.

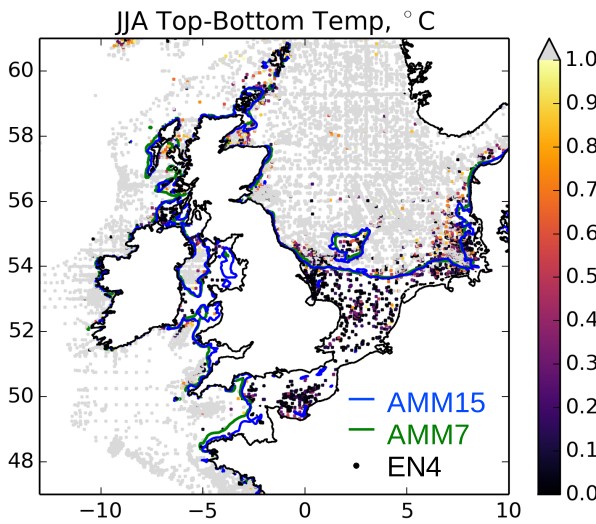

**Figure 5.** Mean summer stratification, indicated by top-bottom temperature difference [°C]. Blue and green lines contour regions with a mean top-bottom temperature difference of 0.5°C in AMM15 and AMM7, respectively. Model results show the seasonal mean (JJA) for 1991-2010, indicating location of seasonal tidal mixing fronts. Shading shows the observed temperature difference (top-bottom), from all monthly-mean EN4 profiles during 1991-2010 (Good et al., 2013). Points showing $> 1$°C are coloured grey for clarity.

### 3.3 Seasonal Stratification

With the onset of stratification in spring-summer, tidal mixing fronts form a key part of the shelf hydrography. The position of these fronts is dependent on the balance between tidal energy and strength of stratification. Assuming a uniform rate of heat input, the location of the fronts is then shown to be dependent on the tidal velocity and depth of the water column
(Simpson and Hunter, 1974). Figure 5 shows the location of tidal mixing fronts in AMM15 and AMM7, compared with observed stratification. This shows that across the majority of the shelf, the fronts are found in a similar location in both models, and compare well with observations. Similarity between the models is consistent with the fact that both have similar representations of the major tidal constituents, and have similar vertical mixing schemes. However, there are improvements in the position of fronts in the western Channel, as well as the west coast of Scotland. This is consistent with the reduced amplitudes
(and hence reduced errors) of M2 seen in Figure 2. Aside from improved representation of the coastline in AMM15, there are also differences between the bathymetry used in AMM15 (EMODnet) and AMM7 (NOOS). In particular, there is an average increase in water column depth off the west coast of Scotland, of the order of 20 m. Partly this may be due to the use of a more recent, improved bathymetric product, based on increased number of available observations. The increased resolution will also allow deep channels between islands to begin to be resolved. This increased depth can then prevent the water column from
being fully mixed during the summer months.

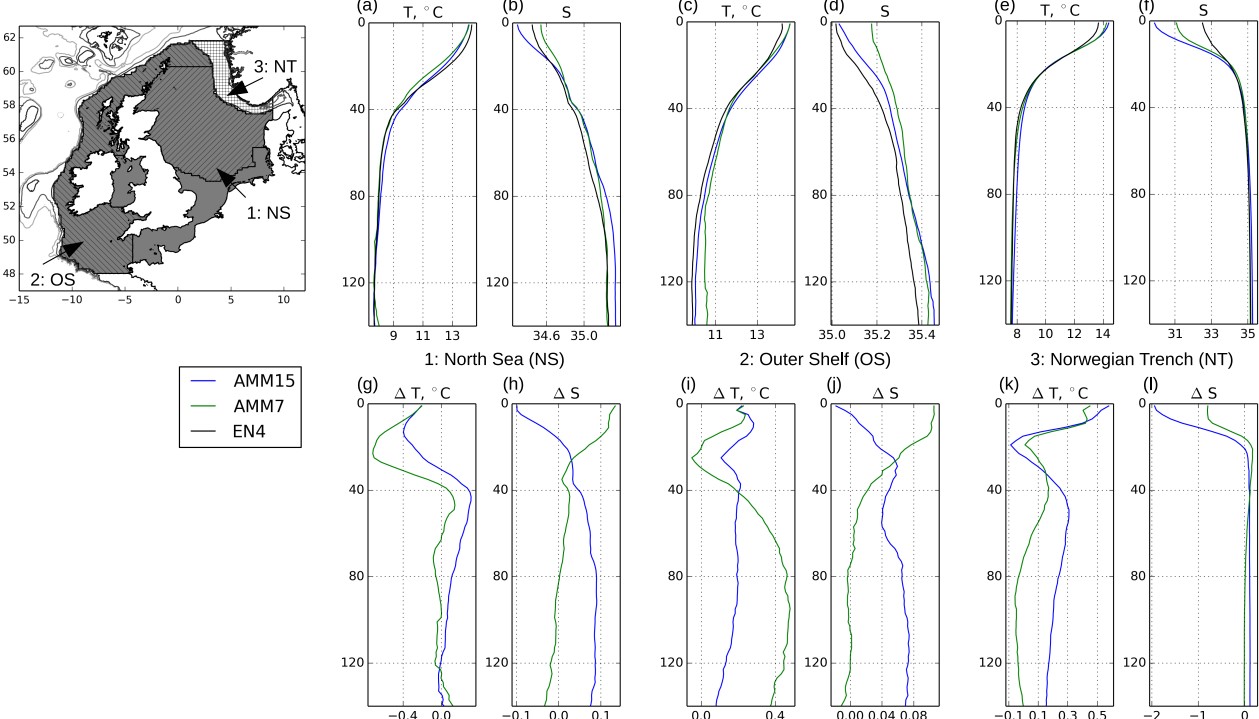

**Figure 6.** Mean summer (JJA) temperature [°C] and salinity profiles for three stratified regions, shown as hatched regions in the upper left panel: North Sea (NS), Outer Shelf (OS) and Norwegian Trench (NT). All panels show 20 year-mean profiles, for JJA, 1991-2010. Observations (black) are monthly EN4 profiles (Good et al., 2013). Upper panels (a-f) show mean profiles with depth, lower panels (g-l) show anomalies with depth for respective profiles, where $\Delta T = \overline{(T_{AMM} - T_{EN4})}$. Results from AMM15 and AMM7 are shown in blue and green, respectively.

Figure 6 shows mean vertical profiles for temperature and salinity during summer for stratified regions across the continental shelf. In the North Sea (Region 1), there is a cool bias at the surface along with a warm bias at depth (Figure 6a,g). There are a number of factors that could influence anomalies across the shelf, including errors in surface fluxes, or advection into or out of the region. Vertical profiles will also be strongly influenced by vertical mixing and light attenuation schemes. While the horizontal resolution has been increased in AMM15, there has been little change in the vertical resolution or parameterisation schemes. Therefore it is unsurprising that similar biases remain in the vertical profiles and stratification, as indicated by a similar surface bias in the region (Figure 3). The warm anomaly at depth during the summer (Figure 6g) will contribute to a warm surface bias during autumn, following the breakdown of stratification (Figure 3d,h).

Contrary to the North Sea, the outer shelf (Region 2, Figure 6c,i) shows a surface which is too warm. This may be related to the warm surface bias that does still exist along the shelf break (Figure 3c), due to a lack of vertical mixing in this region.

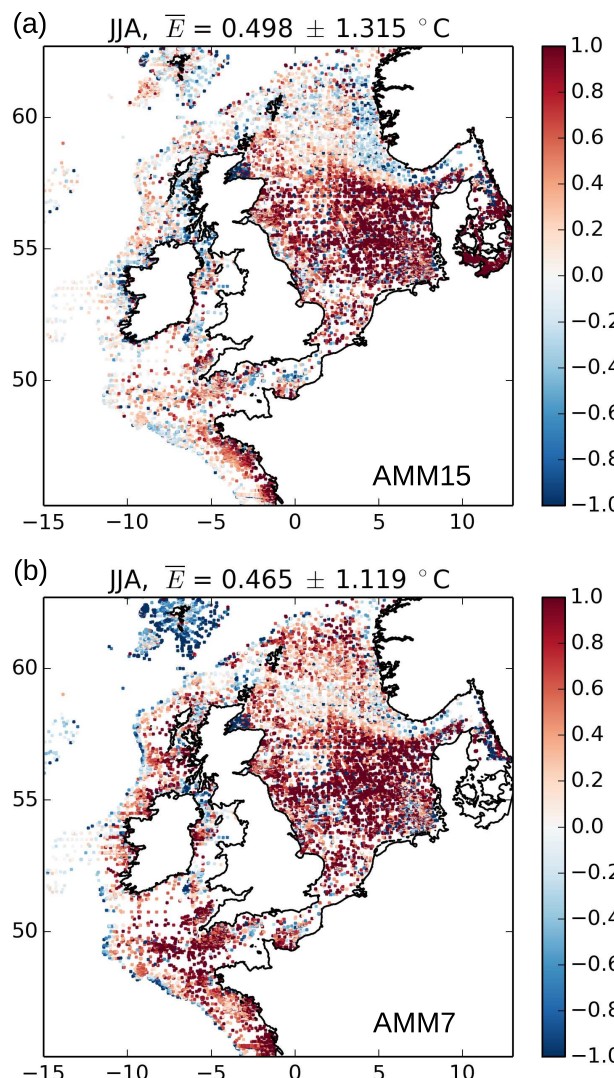

**Figure 7.** Mean seasonal bottom temperature anomalies for model minus observations [°C]. Both panels show 20 year-mean seasonal anomalies, for summer (JJA), 1991-2010. Observations used are monthly EN4 profiles (Good et al., 2013). Upper panel (a) shows anomalies for AMM15-EN4, lower panel (b) shows anomalies for AMM7-EN4. Mean errors ($\overline{E}$) are calculated for the AMM15 domain region, where bathymetry $< 500\,\mathrm{m}$.

Comparison with salinity profiles confirms that the surface is too fresh, whereas the deeper ocean is more saline than observed (Figure 6d,j). For AMM15, the warm bias decreases with depth, with reduced bias compared to AMM7 (Figure 6i).

Figure 7 shows the summer bottom temperature anomalies for both AMM15 and AMM7, compared with EN4 observations. This demonstrates that both models have a warm bias throughout the North Sea. However, anomalies in bottom temperature vary spatially. The mean profiles for the North Sea and outer shelf (Figure 6g,i) show a warm anomaly at depth, consistent with the mean bias shown in Figure 7. However, along the shelf break, AMM7 has a cold bias in bottom temperatures, consistent with a lack of vertical mixing. It is also worth noting that since the depth across the shelf varies (from $\sim 20 - 200\,\mathrm{m}$), the anomalies shown in bottom temperature will not necessarily correspond to the base of the mean vertical profiles shown. For example, Figure 6g shows a maximum temperature anomaly in AMM15 at $40 - 50\,\mathrm{m}$. The largest anomalies in Figure 7 are found towards the shallower southern North Sea and coastal regions (Figure 1).

For AMM15, the bias in bottom temperature is reduced approaching the shelf break (Figures 6i and 7). This suggests that in regions with a greater influence from the open ocean, AMM15 performs better than the current configuration. This may be a result of AMM15 having improved representation of shelf-break processes, or reduced off-shelf biases, which would both influence biases in this region. The mean bias ($\overline{\overline{E}}$) shown in Figure 7 does not appear reduced at higher resolution. However, this includes an increased warm bias in the Baltic for AMM15, outside the AMM7 domain. Excluding these points outside the AMM7 domain, AMM15 is then shown to have a reduced bias compared to AMM7, of $0.366 \pm 1.001$°C compared to $0.465 \pm 1.119$°C, respectively.

Overall, while there have been some improvements in AMM15, similar biases remain in stratification across the shelf. Given that both models have the same number of vertical levels, vertical mixing schemes, and surface forcing, this result is not entirely surprising. Across large areas of the shelf, the climate will be predominantly driven by a balance of vertical forces (surface buoyancy fluxes and vertical mixing) rather than horizontal advection. It is therefore clear that further work is needed to improve the representation of these vertical processes. However, given that there are spatially varying anomalies across the shelf, the response to altering available parameters will vary. Improving the choice of vertical mixing schemes is still an active topic of research (Luneva et al., 2017), and the aim would be to improve those used in future operational systems.

Previous studies have assessed the impact on stratification of using an alternative light attenuation scheme (O'Dea et al., 2017). The uniform RGB scheme used here assumes a Chlorophyll concentration of $0.05\,\mathrm{mg.Chl\,m}^{-3}$ (Lengaigne et al., 2007). This may be appropriate for the majority of the open ocean, but will underestimate chlorophyll concentration throughout this domain. This also neglects additional impact of suspended sediment. The scheme tested by (O'Dea et al., 2017) uses a single-band light attenuation scheme, where the depth of penetration varies with the depth of bathymetry. While this scheme may be appropriate for regions of the North Sea where depth is likely proportional to the water clarity, it does not account for high chlorophyll concentrations in deeper, nutrient-rich waters, such as the Norwegian Trench, and the North East Atlantic. A test has been run using this scheme in the AMM15 domain (not shown). Some improvement is seen in the North Sea, however other regions see increased biases emerge in the summer. A cold surface bias results off-shelf, and SST is also further reduced in the Norwegian Trench, where a cool bias already exists in the summer. Further tests are needed to investigate the impact of including 2D chlorophyll variability, or KD-490 schemes.

The Norwegian Trench shows increased anomalies in AMM15 compared with AMM7 (Region 3, Figure 6k,l). In addition to the fresh anomaly found at the surface (Figure 4), there is also a warm, saline anomaly at depth. These anomalies suggest

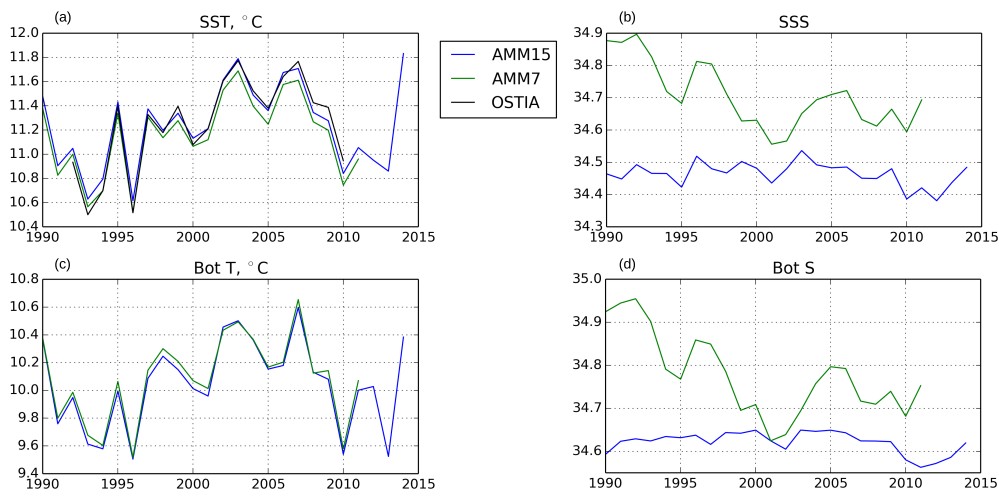

**Figure 8.** Annual mean temperature (left) and salinity (right) for the surface (top) and bottom (bottom) over the continental shelf (shaded region shown in Figure 6). Blue and green lines show mean values for AMM15 and AMM7, respectively. For SST (top left), OSTIA CCI reanalysis (Merchant et al., 2014) are provided for comparison.

a potential difference in the balance of heat and freshwater transport between the Atlantic and Baltic Sea through the Trench. Given that both the Atlantic and Baltic boundaries have been altered in this configuration, the impact of such changes on the Norwegian Trench transport should be the subject of further study. While the addition of barotropic forcing at the Baltic boundary should lead to improvements in AMM15, $1.5\,\mathrm{km}$ resolution is still relatively coarse when compared to narrow channels within the Danish Straits. It is also possible that the difference in SSH forcing used at the Atlantic and Baltic boundaries could lead to an inaccurate flow through the region (e.g., Mattsson, 1996). Further work is needed to assess whether the anomalies seen in AMM15 result from limitations in the model grid and bathymetry, or forcing at either the Baltic or Atlantic boundaries.

### 3.4 Temporal Variability

Both AMM7 and previous configurations have been used for long term climate studies, as well as operational forecasts. Aside from being able to reproduce a mean climatology, it's then also crucial to assess whether model simulations are stable, and can reproduce observed variability in the region. Figure 8 shows the temperature and salinity variability over the shelf during the course of the simulation. For both the models shown here, the surface temperature trends agree with OSTIA data, with an increase through the 1990s reaching a maximum in the mid 2000s, followed by cooling in 2010. Previous studies have shown a warming trend since the 1980s across the NW Shelf, with an average increase in SST of between 0.1 and $0.5°\mathrm{C\,decade}^{-1}$ over the period 1983-2012 (Dye et al., 2013a). This warming has been mostly attributed to atmospheric temperatures (e.g., Meyer et al., 2011; Holt et al., 2012).

Across the shelf, both models show the same variability, consistent with the fact that both are forced with the same atmospheric data (ERA-Interim). However, the mean surface temperature in AMM15 has a reduced bias compared with AMM7.

Analysis of the monthly timeseries (not shown) shows that the difference between the two models is greatest in spring, when AMM7 has a larger cool bias across the shelf (also shown in Figure 3). Breaking the variability down into subregions of the shelf, again both model show similar variability (not shown), with any remaining bias matching that shown in the mean climatology.

Observations from bottom trawl surveys within the North Sea suggest that bottom temperatures have similarly increased, by $\sim 0.2-0.5°\text{C}\,\text{decade}^{-1}$ during 1983-2012 (Dye et al., 2013a). Figure 8 shows the average bottom temperature across the shelf for both AMM15 and AMM7. Both models show similar variability to the surface temperature, increasing from the mid 1990s to a maximum mean temperature in 2007, followed by a decrease in 2010.

It may be expected that the SSS or sub-surface salinity may show greater differences between the models. Temperature on
the shelf is predominantly influenced by surface heat fluxes. While salinity will be partly influenced by evaporation (and hence temperature), it will also be significantly influenced by local river runoff and advection (both of which will differ between the models). Comparing the two models, there is an obvious decreasing trend in AMM7, compared with no significant trend for AMM15 (Figure 8). Similar trends are again found in both the surface and bottom of the water column.

Unfortunately, there are no shelf-wide salinity timeseries for comparison. Previous studies have analysed trends in the
UK coastal waters (Dye et al., 2013b). However, the significance of these trends varies between locations, with river runoff contributing to large interannual variability. It is unclear what has caused the trend in AMM7. Therefore, while AMM15 may appear to be more stable, we cannot say for certain which simulation is more accurate in terms of salinity variability at this time.

## 4   Discussions and Future Work

The next generation ocean forecast model for the European NWS has been introduced here, with the intention that it will become operational in 2018. The new configuration has increased resolution, with $1.5\,\text{km}$ grid spacing throughout the domain, compared to $\sim 7\,\text{km}$ in the previous configuration. A 30 year non-assimilative run has been used to demonstrate the ability of this new configuration (AMM15) to represent the mean state and variability of the region, in comparison with the the current operational system (AMM7).

The increased resolution does make this model more expensive to run. However, the capacity is there to provide this new system, and with increased resolution there is greater potential for added value to end users. The full operational system will use different boundary conditions and include data assimilation. Therefore, it is not possible to say for certain how the skill of operational forecasts will compare with the existing system. However, this study provides insight into how the physics-only configuration performs, and where we should expect to see improvements compared to the existing $7\,\text{km}$ domain. While some
biases are common between the two models, there is an overall improvement in mean climate across the North West Shelf, and there is plenty of scope for further improvement.

Tidal signal within a regional model configuration is to a large extent determined by the boundary conditions and bathymetry. AMM15 and AMM7 have both different bathymetry and tidal forcing at the open boundary. Given this significant change in

configuration, it is then reassuring to see that AMM15 continues to provide a reasonable representation of the major tidal constituents. The minimum depth within the model remains a limiting factor here, so future improvements will focus on the addition of wetting and drying within the domain, which is currently in development for NEMO vn4.0.

Similar biases remain for stratification across the continental shelf, particularly in the North Sea. Given the fact that climate on the shelf can be predominantly driven by a balance of vertical forces (surface buoyancy fluxes and vertical mixing) rather than horizontal advection, it is not surprising that the two models are similar. Both have the same atmospheric forcing, vertical mixing schemes and vertical resolution.

For regions that show little or no improvement, this provides motivation for targeted bias reduction. In the North Sea, there is a need for improved understanding of stratification variability, and how this is represented across the shelf. Bias reduction here will initially focus on improvements to the light attenuation and vertical mixing parameterisation schemes. These schemes should lead to improved stratification and surface climatology across the whole domain, and will be the focus of future study.

There has been substantial progress in developing mixing models in shelf seas over recent decades (e.g. Umlauf and Burchard, 2005), however they still struggle through a lack of specific physical process representation (Luneva et al., 2017). Bringing together recent developments in direct observations of turbulent properties and LES modelling, for example in research projects such as PycnMix (Pycnocline Mixing in shelf seas) and OSMOSIS (Ocean Surface Mixing , Ocean Sub-mesoscale Interaction Study) (Belcher et al., 2012), has the potential to lead to substantial improvements in vertical mixing schemes for the shelf seas.

Further work is also needed to assess currents and transport within the region, along with their impact on model hydrography. In the Norwegian Trench, biases are found to be larger than the current operational system. Heat and freshwater transport through the Trench will be influenced by both the Baltic and Atlantic boundaries. Given the number of factors which are likely to impact on changes seen here (including both the location and data used for boundary forcing), further experiments are needed to assess the response to individual perturbations. In particular, significant changes have been made to the Baltic boundary, which warrant further investigation. Attribution of biases to changes in the location of boundaries, chosen forcing products, or local heat or freshwater fluxes within the region, could then inform future development of the operational system.

This model has been developed with operational implementation as the primary goal. However, aside from this purpose, this configuration also provides an excellent new tool for research. This study has focused on the long term climatology and stability of the model, but there are many differences to be seen on shorter timescales, and smaller spatial scales. As with the Norwegian Trench, further research is needed to attribute improvements in the model climatology across the region to changes in horizontal resolution as opposed to boundary locations, forcing or parameterisation schemes. While small scale processes have not been analysed within AMM15 here, previous studies have already demonstrated the ability of similar resolution models to resolve processes such as eddies, fronts and internal waves across the shelf (e.g., Guihou et al., 2017; Badin et al., 2009; Holt and Proctor, 2008). Further work will involve targeted forecast verification, and assessment of the impact of such processes on forecast skill. There is also a wide scope for process studies here, for example assessing the impact of increased resolution on shelf break exchange, or future climate variability across the shelf.

**Table 2.** Compilation keys for AMM15 simulations.

| Key | Description |
| --- | --- |
| key_bdy | Use open lateral boundaries |
| key_dynspg_ts | Free surface volume with time splitting |
| key_ldfslp | Rotation of lateral mixing tensor |
| key_tide | Activate tidal potential forcing |
| key_vvl | Variable Volume Layer |
| key_zdfgls | Generic Length Scale turbulence scheme |
| key_harm_ana | Restartable tidal analysis |
| key_shelf | Diagnostic switch for output |
| key_iomput | Input output manager |
| key_nosignedzero | Ensure reproducibility with SIGN function |
| key_vectopt_loop | Vector optimisation |

One of the biggest challenges ahead will be to see how the high resolution simulation responds to data assimilation and coupling with biogeochemistry, as part of the operational system. However, this configuration has already been implemented as the ocean component of the UK Environmental Prediction (UKEP) system (Lewis et al., 2017), where it has been coupled with atmospheric and wave models. Initial results are very promising, and demonstrate the value of increased ocean resolution
for simulating the wider climate system.

## 5   Code availability

AMM15 is a regional configuration of NEMO (Nucleus for European Models of the Ocean), at version 3.6 stable (Madec, 2016). Model code is freely available from the NEMO website (www.nemo-ocean.eu). After registration the FORTRAN code is readily available using the open source subversion software (http://subervsion.apache.org). Additional modifications to the
10 NEMO v3.6 trunk are required for AMM15 simulations, and these changes can be found in the NEMO repository. The simulations discussed here were compiled at NEMO r5549. However, the original changes have now been merged under r6232, and can be found within the following branch: branches/UKMO/AMM15_v3_6_STABLE_package. Tests have confirmed that there is no significant difference in model results between these two code revisions.

The compilation keys required for these simulations are listed in Table 2.
An example namelist for the control simulation, containing all chosen parameterisations, can be found under the following DOI: 10.13140/RG.2.2.27237.40164 (Graham, 2017).

## 6 Data availability

The nature of the 4D data generated requires a large tape storage facility. The data that comprise the AMM15 hindcast simulation are of the order of 90TB. However, the data can be made available upon contacting the authors.

Bathymetry was obtained from the EMODnet Portal: EMODnet Bathymetry Consortium, EMODnet Digital Bathymetry (DTM), EMODnet Bathymetry (September 2015 release).

River gauge data was provided by pers. comm from Dr. S. M. van Leeuwen, CEFAS, Lowestoft, UK. The riverine forcing used for this control simulation can be made available upon request.

*Acknowledgements.* We would like to thank the editor and anonymous reviewers for their constructive comments, which helped to improve this manuscript.

Simulations were carried out on the Cray HPC at the Met Office, UK. We acknowledge the Centre for Environmental Data Analysis (CEDA) for the use of JASMIN (Lawrence et al., 2013) for processing model input data.

Funding support is gratefully acknowledged from the Ministry of Defense, the Public Weather Service, and from the Copernicus Marine Environment Monitoring Service.

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
