# Peer review of "AMM15: A new high resolution NEMO configuration for operational simulation of the European North West Shelf"

_Geoscientific Model Development, 2017_

## Referee Comment (RC1) · Anonymous Referee #1 · 26 Jul 2017

The paper at hand, "AMM15: A new high resolution NEMO configuration for operational simulation of the European North West Shelf" describes in some detail the benefit from using an eddy-resolving model of the entire shelf, rather than just in limited, usually near-shore regions. The upgrade from the former system is not limited to a horizontal scale change, as everything except atmospheric forcing, vertical mixing and vertical resolution, has undergone a revision. A practical and very sensible choice.

The paper is excellently written. I think it would benefit from just a few points of clarification.

Comments.

p.5 line 5. Is the tidal forcing an open boundary condition or applied in the entire domain as a tidal potential? Not clear exactly how this is implemented. The paper says, "The amplitude and phase is provided for 12 tidal constituents (surface height and velocity)" . Does NEMO convert this TPX input to forcing terms?

p.5 line 20-27. Not entirely clear if the river data is applied as an annual climatology (one single average value per river) or daily climatology (the annual cycle included). I assume the reason for not using time series rather than climatology is that this was not available at the time.

p.5 line 28. Please state the horizontal resolution of the atmospheric forcing. Only time resolution is given.

p.6 line 14. Are you stating that the use of river climatology based on gauge data is to be prefered over runoff time series based on a hydrological model?

p.12 line 1 and p.18 line 22. The (increased) fresh water bias in the Norwegian Trench is probably rightly connected with the Balitic boundary, which in AMM15 lies south of the Belt Sea. Is there any indication of the correctness of the modelled transport thru the Danish Straits? A large error could exist, possibly stemming from the not very realistic (in this region) 10m minimum depth. How does the model Belt Sea cross-section area, net water transport out of the Baltic, and fresh water export, compare with reality? It is also possible that the boundary data (from Gräwe et al.) is too fresh. This is one instance where the AMM15 high resolution model in some respects is described as leading to worse results than the previous 7km model. It could well be examined just a bit further, not necessarily by further experiments, as stated in the Conclusion. but using the data at hand. But that is just a suggestion.

p.12 line 8. How large is the fresh water reduction, in %?

p. 13 Fig. 5. Some of the graphics is white and thus invisible.

Typos:

p.4 line 14. Kattergat -> Kattegat

p.16 line 6. uses of a -> uses a

---

## Referee Comment (RC3) · Anonymous Referee #3 · 31 Aug 2017

This paper describes the development of the new UK operational forecast model for North West European shelf seas (AMM15). This model will be preplacing the current operational model (AMM7). The predominate change is the horizontal grid resolution, increasing to 1.5 km from 7 km, enabling finer scale processes (such as mesoscale eddies) to be resolved. The paper is very well written but could benefit from some clarifications (listed below). I did struggle with a number of the figures when using a print out. Going back to the PDF and zooming in was helpful and enabled the grey to be distinguished from the white background. Whist this isn't critical – i.e. it's an online journal – some thought could be given to making the figures clearer. The model is described as being the next generation ocean forecast model. Some introduction as to what this

means exactly would be useful, e.g., is the model run weekly/daily and how long a forecast is simulated? The model must also be more computationally demanding that its predecessor, AMM7, and there must be computational considerations when operationalising it. Further to this, the paper describes a set of hind cast simulations. Would operationalizing the model involve using different forcing data? A short discussion on this would be interesting.

P. 5 lines 20-26: I assume that a time series of freshwater flux/discharge was specified for each river? This could be made clearer. In addition, was river temperature and salinity time series used, or if not what values were used or assumptions made? Were daily/monthly averages used and/or what temporal resolution was used?

P. 6 line 24: RMSE is not defined.

Figure 2: The co-tidal charts are quite hard to read, especially on paper/print out. The amplitude is OK, and having a discrete colour scale is helpful, but the phases (white to black) do not seem to equate to the colour scale (grey to white). It's also hard to see the black phase lines on the blue background. In the lower panels (c – f) There is a lot of overlapping observational data points which makes these hard to read. It's hard to know how exactly to make this clearer apart from making the figure larger. This figure could be split (a, b) and (c – f) allowing them all to be larger/clearer.

P. 7: The text says that the amplitude of the M2 tide has reduced errors off the west coast of Scotland. Where exactly do you mean, i.e. out beyond the Outer Hebrides or at the coast (Mull of Kintyre)?

P. 9 line 10: It would be helpful if how the SST anomalies were calculated was explained in the text.

Figure 3 and supporting article text: DJF, MAM, JJA and SON should probably be defined (probably in the text as this is where these abbreviations are used and maybe simply as what season they are). Also, the text refers to season by name in many

cases (e.g. P. 10 line 21) and whist everyone knows by spring you mean MAM, this should also be defined. This could all be done early on by saying seasonal means were calculated for winter (DJF), spring (MAM), and so on. . .

---

## Referee Comment (RC4) · Anonymous Referee #4 · 14 Sep 2017

Summary: The paper presents results from an old and new setup of NEMO, covering the Northwest shelf region (NWS). It includes a model inter-comparison and validation against observations. The new model configuration will replace the previous setup in the operational model setup at the UK MetOffice. The paper focuses on results based on seasonal and yearly calculations. The results are well presented and it clearly illustrates that the new high resolution setup delivers better salinity and temperature results on seasonal to climatic time scales. Furthermore, tidal signal is also better described in the new setup. Model developments and validation are too rarely presented in journals, and it is very interesting to read model development at operational centers. The manuscript discusses processes near the shelf edge, which could be benefitted by the scientific community. Throughout the manuscript results from three regions are presented: 1) Outer shelf, shelf and Norwegian trench. It would be nice if the results were presented in this order for all sections. Sometimes the Norwegian section comes second and sometimes it comes last.

Central issues The authors should present which operational products are produced based on results from the operational model. If products are related to storm surge events, search and rescue or other weather related issues, then this manuscript lacks validation on these phenomenons. It could be analyzing peak error on sea level or presenting results from a storm surge event. For an operational model, the results on time scales from hours to weeks are also important, especially on sea level. Excluding sea level variations from the papers makes it impossible to know if the barotropic transports are sufficiently well simulated, especially if (as the author states) S/T-climate are mainly governed by vertical processes. Still, the (non-tidal) barotropic signal has effect on the state of the North Sea on both short and longer time scales. One main conclusion presented in Section 4 is: Abstract: "Since there has been no change to the vertical resolution or parameterization schemes, performance improvements are not expected in regions where stratification is dominated by vertical processes, rather than advection."

It seems like that this conclusion is based on pp13, line 2-6, thus rather short for being such a central parts in conclusions/abstract and I believe it deserves more attention in Section 3

Minor comments pp 4, line 28. Why is the minimum depth spec. to 10m? Would it not be better to set minimum depth to much less, maybe 3m, and locally increase the depth to at least 10m in Bristol Channel and Gulf of St. Malo? Considor to include a comment on this in the manuscript. pp5, line 18. Method for Tidal forcing is only valid

for AMM15. Method for Tidal forcing for AMM7 is described on pp6 line 16. Merge these two sections into one, preferably at pp.5. pp 6, line 22-23. There are shifts in the position of two amphidromes. Please comment if that is good or bad. pp. 6, line 32. Add English to "Channel". Not all may come to think of English Channel when just writing Channel. Caption to Figure 2 and Table 1. Refer to data source in text also (only in fig caption is not enough). To my knowledge, non-British tide gauge data (e.g. Danish, Norwegian, and German) is not available at BODC. Figure 2. Label and add units to the vertical bars to the right of panels. I do not see the meaning of the phase-bar. For example, it is not possible to differ between phase=80 and phase=240. Comment to pp 8, line 6. Tidal signal downstream of shallow regions will also be affected by too deep minimum depths. Please comment on downstream consequences. pp9, line 12, Are OSTIA observations skin or bulk temperature? Are there any problems validating this observed temperature to a 10m thick model surface layer level? Please comment on that, especially for regions that have seasons with expected shallow surface layers. Maybe some regions do have thicker than 10m for all seasons, others not. I think addressing this will help the reader to interpret the validation better (also related to next three comments). pp 10, line 28-31. Thermal inertia in model during summer that causes a delay in summer heating, and a delay in cooling during early fall (when warm surface layer is being developed and maintained) is , to at least some extent, explained by a too thick surface layer (=10m). pp10, line 25-35. Hypothesis that too warm SST during spring due to too shallow mixed layer depth could be verified by producing a MAM and SON figure similar to Fig6a. Consider to include that analysis. pp12, line 33. Too cold surface water may be explained by the 10 m thick surface layer. This could be investigated by computing the evolution of a single column model with different vertical resolutions, but with same forcing and initial conditions, but this is just a suggestion. Fig. 8 and text pp. 17, line: 5-15: These two paragraphs do not add much information to the study; more than that AMM7 has larger interannual variability. But it cannot be verified which one of these setups does the best job. The observations referred to, cover twice the simulation period and are presented

rather vaguely with only sign of trend (positive) but no rate of change or mean value for the period. If there is a trend, then stability in AMM15 is not reassuring. It should drift. Furthermore, if AMM15 simulates the long term mean value rather good, the decrease in AMM17 until early 2000s may be a good thing, and maybe the increase after 2003 could just as well agree very well with the observed long term trend. My conclusions based on your results may be considered rather speculative, but that is exactly my point. From the interannual salinity data presented in this paper, the authors' results are speculative. More results are needed, or the paragraph should be rewritten or deleted.

Title of section 4 is "Discussion and Conclusions": I do not detect so many conclusions, more discussion and future work. It would be nice to highlight conclusions further in text or add some conclusions drawn from the paper; alternatively rename Section 4 to "Disccusions and Future work". The only obvious conclusion to me is: Section 4: pp 18, line 9-11 "Given the fact that climate on the shelf can be predominantly driven by a balance of vertical forces (surface buoyancy fluxes and vertical mixing) rather than horizontal advection, it is not surprising that the two models are similar. Both have the same atmospheric forcing, vertical mixing schemes and vertical resolution." This is very interesting and I think it should be more clearly presented in Section 3. Vertical processes may govern the salinity/temperature climate in the North Sea, but advection may very well a dominating factor during storm surge events. Section 4: There are no conclusions and/or discussions from tidal section. In fact, it took me a while to realize that it is only baroclinic features are discussed. Please add a paragraph of the tidal-results presented in 3.1. It could be something like: Tidal signal within a model setup covering the North Sea is to a large extent determined by the boundary conditions and bathymetry. AMM15 and AMM7 have different bathymetry and tidal forcing at the open boundary. It seems like this step is larger than comparing two different models (e.g. NEMO and ROMS) with same bathymetry and tidal forcing (and advections schemes). To me, the tidal part is not considered to be an update (from AMM15 to AMM7), but a replacement to a new setup. Feel free to add something like

this, or choosing some other angle of the tidal-results presented in Section 3.1.

Please also note the supplement to this comment:
https://www.geosci-model-dev-discuss.net/gmd-2017-127/gmd-2017-127-RC4-supplement.pdf

---

## Author Comment (AC1) · 1 Dec 2017

The comment was uploaded in the form of a supplement:
https://www.geosci-model-dev-discuss.net/gmd-2017-127/gmd-2017-127-AC1-
supplement.pdf

---

## Author Response (AR2)

**AMM15: A new high resolution NEMO configuration for operational simulation of the European North West Shelf**

Graham, J. A., O'Dea, E., Holt, J., Polton, J., Hewitt, H. T., Furner, R., Guihou, K., Brereton, A., Arnold, A., Wakelin, S., Castillo Sanchez, J. M., and Mayorga Adame, C. G.

**Response to Editor**

We would like to thank the editor for their time taken to review the revised manuscript. Their comments are listed below, with our responses provided in green text. The revised, marked-up manuscript follows these responses.

Topical Editor's comments

I am happy that the majority of reviewer comments have been addressed and responded to with the revised article. However I would ask you to review the following comments:

Given the emphasis on variability and currents in the introduction, and the specific comments from reviewer 2, you should expand the discussion on future work in this area (page 20 line 33--35), particularly as the focus of this paper was not on the verification of the submesoscale or mesoscale dynamics.

We have now modified this discussion as follows:

*This study has focused on the long term climatology and stability of the model, but there are many differences to be seen on shorter timescales, and smaller spatial scales. … While small scale processes have not been analysed within AMM15 here, previous studies have already demonstrated the ability of similar resolution models to resolve processes such as eddies, fronts and internal waves across the shelf (e.g. Badin et al., 2009; Holt and Proctor, 2008; Guihou et al., 2017). Further work will involve targeted forecast verification, and assessment of the impact of such processes on forecast skill. There is also a wide scope for process studies here, for example assessing the impact of increased resolution on shelf break exchange, or future climate variability across the shelf.*

I still find the description of the initialisation for AMM7 (page 6, lines 26-31) to be unclear. e.g. please clarify "differ" on line 27.

We apologise that this was unclear, and have now clarified the text here as follows:

*The AMM7 hindcast spans 1981-2012, with the method of forcing and initialisation similar to those outlined for AMM15. However, for its forcing and initialisation, AMM7 used earlier versions of both the ORCA025 and GLOSEA configurations, therefore differences can be expected in these forcing products. The free-running ORCA025 simulation was used for initialisation in January 1981, and boundary forcing up to 1990. The remainder of the simulation used boundary conditions from GLOSEA. Although the GLOSEA data comes from an earlier version than that used for AMM15, both versions include data assimilation. Therefore, there should be greater similarity in the boundary*

*conditions for AMM7 and AMM15 from 1990 onwards. Analysis of model climatology will then focus on a common 20 year period in both simulations, 1991-2010.*

I believe reviewer 3 was requesting DJF etc. to be defined explicitly on first use, "(December, January, February)".

Apologies for this omission, each season in now defined in section 3.2.

In your response to reviewer 4 you mention calculations which use a minimum depth of 6m. You could consider mentioning that these were performed, without significant improvements being observed.

We thank the reviewer for this suggestion, and now mention this at the end of Section 3.1:

*Tests were carried out with a reduced minimum depth of 6 m in AMM15. However, this led to no significant improvement in tidal simulations. Wetting and drying is currently under development for NEMO vn4.0 …*

Page 19 lines 10-20: I think you should revisit reviewer 4 comments regarding this section, adding further qualifications, or deleting the text as suggested by reviewer 4.

As mentioned in response to reviewer 4, we feel that it is worth presenting these time series here, for the benefit of future users. However, we understand that the conclusions drawn so far are limited. Based on the editor's comments, we have then updated the text here to acknowledge these limitations.

*Unfortunately, there are no shelf-wide salinity timeseries for comparison. Previous studies have analysed trends in the UK coastal waters (Dye et al., 2013b). However, the significance of these trends varies between locations, with river runoff contributing to large interannual variability. It is unclear what has caused the trend in AMM7. Therefore, while AMM15 may appear to be more stable, we cannot say for certain which simulation is more accurate in terms of salinity variability at this time.*

Page 20 lines 1-6: Since you have followed the reviewer 4 suggestion quite closely, please consider adding an appropriate reviewer acknowledgement.

Apologies for this omission, we have now thanked both the editor and reviewers for their useful comments within the Acknowledgements.

Page 3 line 26: Please clarify "future developments".

This has now been clarified as follows:

*… pending future developments, including data assimilation …*

Page 12 line 9, page 19 line 16: Avoid the contractions "isn't" and "can't".

Done.

[revised manuscript text omitted]